# Mixture Representation Learning with Coupled Autoencoding Agents

## Abstract

Jointly identifying a mixture of discrete and continuous factors of variability can help unravel complex phenomena. We study this problem by proposing an unsupervised framework called coupled mixture VAE (cpl-mixVAE), which utilizes multiple interacting autoencoding agents. The individual agents operate on augmented copies of training samples to learn mixture representations, while being encouraged to reach consensus on the categorical assignments. We provide theoretical justification to motivate the use of a multi-agent framework, and formulate it as a variational inference problem. We benchmark our approach on MNIST and dSprites, achieving state-of-the-art categorical assignments while preserving interpretability of the continuous factors. We then demonstrate the utility of this approach in jointly identifying cell types and type-specific, activity-regulated genes for a single-cell gene expression dataset profiling over 100 cortical neuron types.

## 1 Introduction

Complex phenomena can be attributed to a mixture of discrete and continuous factors of variability. Such complexity is crucial to understand in a variety of different contexts, from learning models for image datasets to identifying factors underlying neuronal identity. A common approach to study these phenomena is clustering, which can produce representations that jointly capture the dependence on discrete and continuous factors. Generative models can learn such representations, which has recently received attention from the deep learning community. Deep Gaussian mixture models are among the first deep generative models to jointly represent discrete and continuous factors, in which a continuous representation is decomposed into discrete clusters (Johnson et al., 2016; Dilokthanakul et al., 2016; Jiang et al., 2017). However, such models have mainly focused on clustering without regard to interpretability. Adversarial and variational methods have been proposed to learn mixture representations that can identify interpretable continuous factors. While adversarial learning, e.g. InfoGAN (Chen et al., 2016) is susceptible to stability issues (Kim & Mnih, 2018; Dupont, 2018; Jeong & Song, 2019), variational approaches, e.g. JointVAE and CascadeVAE have produced promising and more stable results (Dupont, 2018; Jeong & Song, 2019). However, such variational methods utilizing a single autoencoding agent rely either on a heuristic data-dependent embedding capacity, or on solving a separate optimization problem for the discrete variable. Thus, learning interpretable and stable mixture representations remains challenging.

We introduce a multi-agent variational framework to jointly infer discrete and continuous factors through collective decision making, while sidestepping heuristic approaches used by single-agent frameworks. Coupling of autoencoding agents has been previously studied in the context of multi-modal recordings, where each agent learns a continuous latent representation for one of the data modalities (Feng et al., 2014; Gala et al., 2019). Here, we propose pairwise-coupled autoencoders to learn a mixture representation for a single data modality in an unsupervised fashion. Each autoencoder agent receives an augmented copy of the given sample with the same class label. To achieve this, we design a novel type-preserving augmentation that generates noisy copies of the data using within-class variabilities, while preserving its class identity. Coupling across the agents is achieved by encouraging categorical variables to be invariant under the augmentation, which regularizes the agents to learn interpretable representations. We demonstrate that such a coupled multi-agent architecture can increase inference accuracy and robustness by exploiting within-cluster variabilities, without requiring a prior distribution on the relative abundances of categories.

Our contributions can be summarized as follows: (i) We first provide theoretical justification to motivate the advantage of collective decision making for more accurate categorical assignments,

utilizing noisy copies of the same sample. To obtain such samples, we propose an unsupervised type-preserving augmentation method. (ii) We formulate collective decision making as a variational inference problem with multiple agents. In this formulation, we introduce an approximation of Aitchison distance in the simplex to compare categorical assignments of the agents, which avoids mode collapse. (iii) We benchmark our method and display its superiority over comparable approaches using the MNIST and dSprites datasets. (iv) Finally, we apply the method to a challenging single cell gene expression dataset for a population of neurons. We demonstrate that our method can be used to discover discrete categories referred to as neuronal types and type-specific genes regulating the continuous within-type variability (e.g., metabolic states, disease states).

## 2 RELATED WORK

As introduced above, recent studies on joint learning of discrete and continuous factors in generalized mixture models focus on variational or adversarial approaches (Dupont, 2018; Jeong & Song, 2019; Chen et al., 2016). There is also a rich literature that focus on clustering in mixture models and do not attempt to characterize the continuous variability: Dilokthanakul et al. (2016) and Jiang et al. (2017) performed variational inference in mixture models using autoencoding architectures. Tian et al. (2017) applied the alternating direction method of multipliers (ADMM) to use classical clustering algorithms in conjunction with a neural network. Guo et al. (2016) and Locatello et al. (2018b) used gradient boosting approaches (Friedman, 2001) to iteratively fit mixture models in variational frameworks. Moreover, the idea of improving the clustering performance through seeking a consensus across similar *agents* has been explored in both unsupervised (Monti et al., 2003; Kumar & Daumé, 2011) and semi-supervised contexts (Blum & Mitchell, 1998). However, the proposed consensus clustering approach is attempting to learn an interpretable continuous variability. Here, in contrast, beyond joint disentangling, we propose a framework in which the agents seek a consensus, at the time of learning the mixture representation.

While our method does not assume *any* prior/supervising information, the individual agents in our approach can be considered to provide a form of weak supervision to each other. Bouchacourt et al. (2017) demonstrated a multi-level variational autoencoder as a weak supervised disentanglement approach for different factors of variability by both revealing that observations within groups share the same class label where the class label variable takes values from a finite set of labels and are represented by a Gaussian distribution. Recently, Locatello et al. (2020) improved this framework by assuming that observation pairs share at least one underlying factor, and demonstrated disentangling of continuous (but not discrete) factors on image sets.

## 3 PRELIMINARIES

For an observation $\mathbf{x} \in \mathbb{R}^D$, a variational autoencoder (VAE) learns a generative model $p_{\boldsymbol{\theta}}(\mathbf{x}|\mathbf{z})$ and a variational distribution $q_{\boldsymbol{\phi}}(\mathbf{z}|\mathbf{x})$, where $\mathbf{z} \in \mathbb{R}^M$ for $M \ll D$ is a latent variable with a parameterized distribution $p(\mathbf{z})$ (Kingma & Welling, 2013). *Disentangling* different sources of variability into different dimensions of $\mathbf{z}$ enables an interpretable selection of latent factors (Higgins et al., 2017; Locatello et al., 2018a). However, in many real-world applications the inherent mixture distribution of continuous and discrete variations is often overlooked. This problem can be addressed within the VAE framework in an unsupervised fashion by introducing a categorical latent variable $\mathbf{c} \in \mathcal{S}^K$, denoting the class label defined in a $K$-simplex, alongside the continuous latent variable $\mathbf{s} \in \mathbb{R}^M$. Here, we refer to the continuous variable $\mathbf{s}$ as the *state* or *style* variable interchangeably. Assuming $\mathbf{s}$ and $\mathbf{c}$ are independent random variables, the evidence lower bound (ELBO) (Blei et al., 2017) for a single autoencoding agent with the distributions parameterized by $\boldsymbol{\theta}$ and $\boldsymbol{\phi}$ is given by,

$$\mathcal{L}(\boldsymbol{\phi}, \boldsymbol{\theta}) = \mathbb{E}_{q_{\boldsymbol{\phi}}(\mathbf{s},\mathbf{c}|\mathbf{x})} \left[ \log p_{\boldsymbol{\theta}}(\mathbf{x}|\mathbf{s},\mathbf{c}) \right] - D_{KL}\left( q_{\boldsymbol{\phi}}(\mathbf{s}|\mathbf{x}) \| p(\mathbf{s}) \right) - D_{KL}\left( q_{\boldsymbol{\phi}}(\mathbf{c}|\mathbf{x}) \| p(\mathbf{c}) \right). \quad (1)$$

Maximizing ELBO in Eq. 1 to jointly learn $q(\mathbf{s}|\mathbf{x})$ and $q(\mathbf{c}|\mathbf{x})$ is challenging due to the mode collapse problem, where the network ignores a subset of latent variables. Akin to $\beta$-VAE (Higgins et al., 2017; Burgess et al., 2018), JointVAE assigns controlled information capacities to both continuous and categorical factors to prevent mode collapse (Dupont, 2018). A drawback of this method is that the capacities are dataset-dependent, and need to be tuned empirically over training iterations. As an alternative, CascadeVAE (Jeong & Song, 2019) maximizes the ELBO by iterating over two separate optimizations for the continuous and categorical variables after a warm-up period, instead of a fully gradient-based optimization. Although the computational cost for the suggested optimization for the categorical variable has an approximately linear dependence on the number of categories and

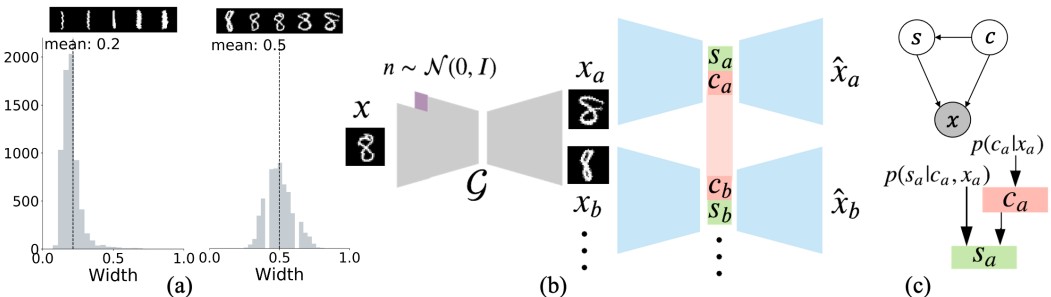

Figure 1: (a) Empirical distributions for the continuous state variable representing stroke width are digit-dependent, illustrating dependence of style on type. (b) Multi-agent autoencoder framework proposed as cpl-mixVAE model. This framework uses a type-preserving augmentation network $\mathcal{G}$. Each augmented sample is obtained by first projecting the sample to a lower dimension, then perturbing it with noise $\mathbf{n}$, and finally projecting it back to the data space. Agents receive noisy copies of given samples $\mathbf{x}$, i.e. $\{\mathbf{x}_a, \mathbf{x}_b, \dots\}$, where they all belong to the same class label, to learn mixture representations, i.e. $\{q(\mathbf{c}_a, \mathbf{s}_a), q(\mathbf{c}_b, \mathbf{s}_b), \dots\}$. Agents cooperate to learn the categorical assignment, $p(\mathbf{c})$. Cooperation is achieved by imposing a penalty on mismatches in the categorical assignments. (c) Each autoencoder agent learns type dependence of the state variable according to the graphical model.

batch size, it can still be a deterrent for problems with numerous categories and unbalanced datasets requiring larger batch sizes. Thus, single-agent VAEs fall short of efficiently learning interpretable mixture representations, either due to their reliance on a heuristic embedding capacity, or lacking a fully variational approach.

In addition to the issues discussed above, the performance and interpretability of those VAE approaches are further limited by the common assumption that the continuous variable representing the style of the data is independent of the class label. In practice, style often depends on class label. For instance, even for the well-studied MNIST dataset, the histograms of common digit styles, e.g. "width", markedly vary for different digits (Fig. 1a). Moreover, in the discussed approaches, e.g. JointVAE, unsupervised clustering of only the continuous variable achieves a relatively high classification accuracy ($\sim 66\%$, see supplementary F and G), underscoring that the independence assumption is not valid.

## 4 COUPLED MIXTURE VAE FRAMEWORK

The key intuition behind multi-agent networks is cooperation for decision making to improve probabilistic estimation. Collective decision making has been studied under different contexts and a popular name referring to its advantages is the "wisdom of the crowd" (Surowiecki, 2005). When unanimous decisions made by a crowd (multiple agents) form a probability distribution, multiple estimates from the same sample increase the expected probability of true assignment. This is theoretically justified by Proposition 1 in the context of categorical decision making.

**Definition 1. (A-agent VAE Framework)** *We define the A-agent VAE as an A-tuple of independent and architecturally identical autoencoding agents, where each agent, e.g. a-th agent, parameterizes a mixture model distribution with $\boldsymbol{\theta}_a, \boldsymbol{\phi}_a$. While each agent has its own mixture representation with potentially non-identical parameters, all agents cooperate to learn $q_{\boldsymbol{\phi}_a}(\mathbf{c}_a|\mathbf{x}_a)$ and $q_{\boldsymbol{\phi}_a}(\mathbf{c}_a)$ via a cost function at the time of training.*

**Definition 2. (Confidence)** *Suppose $\mathbf{x}$ is generated by a multivariate mixture distribution so that $p(\mathbf{x}) = \sum_c p(c)p(\mathbf{x}|c)$, where $p(c)$ denotes an arbitrary prior for the relative abundances of categories. The assignment confidence for category $k$ for samples belonging to category $m$ can be expressed as follows.*

$$\mathcal{C}_m(k) = \mathbb{E}_{\mathbf{x}|m}\left[\log p(c = k|\mathbf{x})\right] \tag{2}$$

**Proposition 1.** *Consider the problem of mixture representation learning in a multi-agent VAE framework with $A \geq 2$ agents using type-preserving data augmentation, where the accuracy of categorical assignment for a single agent is imperfect. The confidence of the correct assignment for the multi-agent VAE is higher than that of the single agent VAE. Moreover, there exists an A such that the correct category receives the highest confidence score in the A-agent framework, independent of the categorical prior.* (Proof in supplementary Section A)

While in an unsupervised framework, defining the required number of agents in the absence of categorical prior and category-dependent information remains a challenge, in the particular case of uniform prior distribution of categories, we have the following Corollary.

**Corollary 1.** *For a uniform prior on the discrete factors, one pair of VAE agents ($A = 2$) is sufficient to increase the confidence of correct categorical assignment.* (see supplementary Section A)

### 4.1 MULTI-AGENT VAE

Using the insight obtained from Proposition 1, we formulate collective decision making for an $A$-agent VAE network (Fig. 1b) as the following constrained optimization.

$$\max \quad \mathcal{L}_{\mathbf{s}_1|\mathbf{c}_1}(\boldsymbol{\phi}_1, \boldsymbol{\theta}_1) + \cdots + \mathcal{L}_{\mathbf{s}_A|\mathbf{c}_A}(\boldsymbol{\phi}_A, \boldsymbol{\theta}_A)$$
$$\text{s.t. } \mathbf{c}_1 = \cdots = \mathbf{c}_A \tag{3}$$

Here, $\mathcal{L}_{\mathbf{s}_a|\mathbf{c}_a}(\boldsymbol{\phi}_a, \boldsymbol{\theta}_a)$ is the variational loss for agent $a$ as follows,

$$\mathcal{L}_{\mathbf{s}_a|\mathbf{c}_a}(\boldsymbol{\phi}_a, \boldsymbol{\theta}_a) = \mathbb{E}_{q(\mathbf{s}_a,\mathbf{c}_a|\mathbf{x})}\left[\log p(\mathbf{x}_a|\mathbf{s}_a, \mathbf{c}_a)\right] - \mathbb{E}_{q(\mathbf{c}_a|\mathbf{x}_a)}\left[D_{KL}\left(q(\mathbf{s}_a|\mathbf{c}_a, \mathbf{x}_a)\|p(\mathbf{s}_a|\mathbf{c}_a)\right)\right]$$
$$- \mathbb{E}_{q(\mathbf{s}_a|\mathbf{c}_a,\mathbf{x}_a)}\left[D_{KL}\left(q(\mathbf{c}_a|\mathbf{x}_a)\|p(\mathbf{c}_a)\right)\right]. \tag{4}$$

In Eq. (4), for each agent, we use the graphical model in Fig. 1c and modify the loss function in Eq. (1) by conditioning state on the categorical variable (derivation in supplementary Section B). Not only is it challenging to solve the maximization in Eq. 3 due to the equality constraint, but the objective remains a function of the prior $p(\mathbf{c})$ which is unknown, and typically non-uniform. To overcome this, we introduce an equivalent formulation for Eq. 3 based on the pairwise coupling paradigm as follows (derivation in supplementary Section C).

$$\max \quad \sum_{a=1}^{A} (A-1)\left(\mathbb{E}_{q(\mathbf{s}_a,\mathbf{c}_a|\mathbf{x}_a)}\left[\log p(\mathbf{x}_a|\mathbf{s}_a, \mathbf{c}_a)\right] - \mathbb{E}_{q(\mathbf{c}_a|\mathbf{x}_a)}\left[D_{KL}\left(q(\mathbf{s}_a|\mathbf{c}_a, \mathbf{x}_a)\|p(\mathbf{s}_a|\mathbf{c}_a)\right)\right]\right)$$
$$- \sum_{a<b} \mathbb{E}_{q(\mathbf{s}_a|\mathbf{c}_a,\mathbf{x}_a)}\mathbb{E}_{q(\mathbf{s}_b|\mathbf{c}_b,\mathbf{x}_b)}\left[D_{KL}\left(q(\mathbf{c}_a|\mathbf{x}_a)q(\mathbf{c}_b|\mathbf{x}_b)\|p(\mathbf{c}_a, \mathbf{c}_b)\right)\right]$$
$$\text{s.t. } \mathbf{c}_a = \mathbf{c}_b \ \forall a, b \in [1, A], \ a < b \tag{5}$$

Here, in the last term, the KL divergence across coupled agents is a function of the joint distribution $p(\mathbf{c}_a, \mathbf{c}_b)$, rather than $p(\mathbf{c})$. We relax Eq. 5 into an unconstrained problem by assuming a differentiable form for $p(\mathbf{c}_a, \mathbf{c}_b)$ (full derivation in supplementary Section D).

$$\max \quad \sum_{a=1}^{A} (A-1)\left(\mathbb{E}_{q(\mathbf{s}_a,\mathbf{c}_a|\mathbf{x}_a)}\left[\log p(\mathbf{x}_a|\mathbf{s}_a, \mathbf{c}_a)\right] - \mathbb{E}_{q(\mathbf{c}_a|\mathbf{x}_a)}\left[D_{KL}\left(q(\mathbf{s}_a|\mathbf{c}_a, \mathbf{x}_a)\|p(\mathbf{s}_a|\mathbf{c}_a)\right)\right]\right)$$
$$+ \sum_{a<b} H(\mathbf{c}_a|\mathbf{x}_a) + H(\mathbf{c}_b|\mathbf{x}_b) - \lambda\mathbb{E}_{q(\mathbf{c}_a|\mathbf{x}_a)}\mathbb{E}_{q(\mathbf{c}_b|\mathbf{x}_b)}\left[d^2(\mathbf{c}_a, \mathbf{c}_b)\right] \tag{6}$$

According to the final expression in Eq. 6, the agents try to achieve identical categorical assignments while independently learning their own style variables. For each pair of agents, there are two entropy based confidence penalty terms, which are mode collapse regularizers (Pereyra et al., 2017). There is also distance $d(\mathbf{c}_a, \mathbf{c}_b)$ between a pair of categorical variables, which encourages the consensus on the categorical assignment controlled by *coupling hyperparameter* $\lambda$. The distance is defined as $d(\mathbf{c}_a, \mathbf{c}_b) = \|clr(\mathbf{c}_a) - clr(\mathbf{c}_b)\|_2, \forall \mathbf{c}_a, \mathbf{c}_b \in \mathcal{S}^K$, where $clr(.)$ denotes the isometric centered-log-ratio transformation and therefore $d$ satisfies the conditions of a mathematical metric according to Aitchison geometry (Aitchison, 1982; Egozcue et al., 2003). To sample from $q(\mathbf{c}_a|\mathbf{x}_a)$ in a gradient descent framework, we use the Gumbel-softmax distribution (Jang et al., 2016; Maddison et al., 2014) with a temperature parameter $0 < \tau \leq 1$.

In the rest of this study, we refer to the model in Eq. 6 as *cpl-mixVAE* (Fig. 1b). Note that this formulation can be easily extended to include an additional hyperparameter to encourage disentanglement of continuous variables as in $\beta$-VAE Higgins et al. (2017). To train this model in an unsupervised fashion according to Eq. 6, we require augmented samples $\mathbf{x_a}$ for any given sample $\mathbf{x}$.

### 4.2 TYPE-PRESERVING AUGMENTATION

Augmentation can be considered as a generative process (Antoniou et al., 2017). We seek a generative model that not only learns the data distribution, but also transformations that represent within-class variations in an unsupervised manner. Learning such transformations is generally not straightforward,

and requires prior knowledge about the underlying invariances. While conventional transformations such as rotation, scaling, or translation can serve as type-preserving augmentations for many image datasets, they may not capture the richness of the underlying process. Moreover, such augmentation strategies cannot be used when within-class invariance are unknown. Suggested alternatives to conventional augmentations either rely on class label, or are specific to image data (Hauberg et al., 2016; Jaiswal et al., 2018; Antoniou et al., 2017).

To this end, inspired by DAGAN (Antoniou et al., 2017), we propose an unsupervised type-preserving augmentation using a VAE-GAN (Larsen et al., 2016)-like architecture, Fig. 1.b. We seek a network $\mathcal{G}$ such that a noisy copy, $\mathbf{x_a}$ can be obtained as a variation of the given sample, $\mathbf{x}$, based on its low dimensional representation that is concatenated with Gaussian noise $\mathbf{n}$. To prevent the network from disregarding the noise, we formulate the training procedure as the following minmax optimization which uses a discriminator network $\mathcal{D}$ as a regularizer.

$$\min_{\mathcal{G}} \max_{\mathcal{D}} \mathcal{V}(\mathcal{D}, \mathcal{G}) - \mathcal{R}(\mathcal{G}) + \mathcal{T}_\alpha(\mathcal{G}) + \gamma d(\mathcal{G}) \tag{7}$$

While training, $\mathcal{G}$ generates two samples: $\mathbf{x_n}$ and $\mathbf{x_{\not{n}}}$. The former denotes $\mathbf{x_a}$, and the latter is a sample generated in the absence of noise. In Eq. 7, $\mathcal{V}(\mathcal{D}, \mathcal{G}) = \mathbb{E}_{\mathbf{x}}[\log \mathcal{D}(\mathbf{x})] + \mathbb{E}_{\mathbf{x}}[\log(1 - \mathcal{D}(\mathbf{x_{\not{n}}}))] + \mathbb{E}_{\mathbf{x,n}}[\log(1 - \mathcal{D}(\mathbf{x_n}))]$ is the value function for the joint training of the discriminator and generator; $\mathcal{R}(\mathcal{G}) = \mathbb{E}_{q(\mathbf{z}|\mathbf{x})}[\log p(\mathbf{x}|\mathbf{z})]$ is the reconstruction loss, which operates only over $\hat{\mathbf{x}}$; $\mathcal{T}_\alpha(\mathcal{G}) = \max(\|\mathbf{x} - \mathbf{x_{\not{n}}}\|_2 - \|\mathbf{x} - \mathbf{x_n}\|_2 + \alpha, 0)$ is the triplet loss that prevents network $\mathcal{G}$ from disregarding noise and generating identical samples; and $d(\mathcal{G}) = D_{KL}(q(\mathbf{z}|\mathbf{x})\|q(\mathbf{z}|\mathbf{x}, \mathbf{n}))$ is the distance between the latent variables in the absence and presence of noise. $d(\mathcal{G})$ is a regularizer to encourage original and noisy samples to be located close to one another in the latent space and is controlled by hyperparameter $\gamma \ll 1$.

### 4.3 Stabilizing the training by mini-batch variance

The solution to the maximization problem in Eq. 6, which includes minimization of $d(\mathbf{c}_a, \mathbf{c}_b)$, has trivial local optima that result in the mode collapse issue (Lucas et al., 2019), in which during learning the network ignores a subset of the discrete latent space. For instance, in the extreme case, the network learns $\mathbf{c}_{a_n} = \mathbf{c}_{b_n} = \mathbf{c}, \forall n$, where $n$ denotes the sample index. In this scenario, the state variable is compelled to act as the latent variable of a classical variational autoencoder, while the model fails to deliver an interpretable mixture representation despite achieving an overall low loss value. We regularize the Aitchison distance $d(\mathbf{c}_{a_n}, \mathbf{c}_{b_n})$, between the categorical assignments of the $n$-th samples of agents $a$ and $b$ by using mini-batch statistics to avoid mode collapse: $d_\sigma^2(\mathbf{c}_{a_n}, \mathbf{c}_{b_n}) = \sum_k \left( \sigma_{a_k}^{-1} \log c_{a_{n_k}} - \sigma_{b_k}^{-1} \log c_{b_{n_k}} \right)^2$, where $\sigma_{a_k}^2$ indicates the variance of the $k$-th category of agent $a$. In the following proposition we show that $d_\sigma^2$ is an approximation of the Aitchison distance in the probability simplex.

**Proposition 2.** *Suppose $\mathbf{c}_{a_n}, \mathbf{c}_{b_n} \in \mathcal{S}^K$, where $\mathcal{S}^K$ is a simplex of $K$ parts and $n$ is the sample index. If $d(\mathbf{c}_{a_n}, \mathbf{c}_{b_n})$ denotes the Aitchison distance, then*

$$d_\sigma^2(\mathbf{c}_{a_n}, \mathbf{c}_{b_n}) - d^2(\mathbf{c}_{a_n}, \mathbf{c}_{b_n}) \leq \frac{1}{K}\left( (\tau_\mathbf{c} + \tau_\sigma)^2 + K^2\tau_\sigma^2 - \Delta_\sigma^2 \right)$$

*where $\tau_\mathbf{c} = \max\limits_k \{\log c_{a_{n_k}} - \log c_{b_{n_k}}\}$, $\tau_\sigma = \max\limits_k \{(\sigma_{a_k}^{-1} - 1)\log c_{a_{n_k}} - (\sigma_{b_k}^{-1} - 1)\log c_{b_{n_k}}\}$, and $\Delta_\sigma = \sum\limits_k (\sigma_{a_k}^{-1} - 1)\log c_{a_{n_k}} - (\sigma_{b_k}^{-1} - 1)\log c_{b_{n_k}}$.* (Proof in supplementary Section E)

Accordingly, as the Gumbel-softmax approximations of the categorical variable of the agents move closer to each other on the simplex, $d_\sigma$ converges to $d$.

## 5 Experiments

We assess the performance of cpl-mixVAE for three different datasets. To facilitate comparisons with other methods, first we conducted experiments on two benchmark datasets: MNIST and dSprites. Additionally, we used a single cell RNA-sequencing dataset (scRNA-seq) (Tasic et al., 2018), to evaluate the utility of our approach in identifying neuronal cell types and type-specific genes regulating the continuous within-type variability.

**MNIST**: According to the approximately uniform distribution of handwritten digits in the dataset, we used a 2-agent cpl-mixVAE with shared categorical variable to learn an interpretable representation.

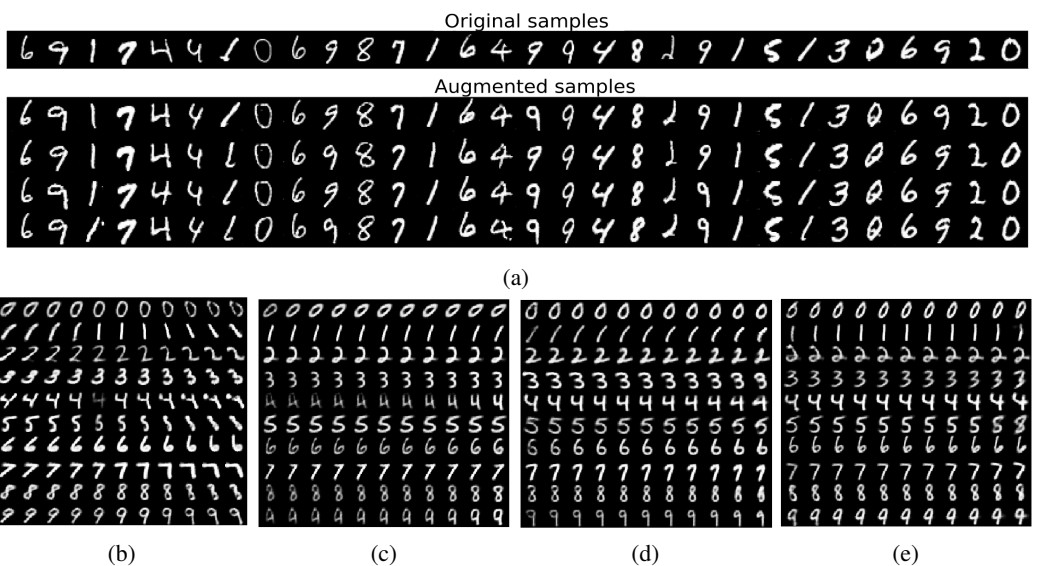

Figure 2: (a) Augmented samples for the MNIST dataset generated by the type-preserving augmentation conserve type of the original sample. (b-e) Continuous latent traversals of $1^{st}$ agent of the cpl-mixVAE framework with two autoencoding agents, where each agent mixture representation is parameterized with 10-dimensional continuous and 10-dimensional categorical variables. Examples of (b) rotation angle, (c) stroke thickness, (d) character width and (e) roundness of looped features are presented. The discrete variable $\mathbf{c}$ is constant for all reconstructions in the same row.

Each agent learned a mixture generative model including a 10-dimensional categorical variable representing digits (type), and a 10-dimensional continuous random variable representing the writing style (state). To generate noisy samples, the augmenter was trained on the MNIST dataset ahead of training the cpl-mixVAE model. Fig. 2a displays example noisy samples generated by the type-preserving augmentation for MNIST. To quantitatively evaluate the proposed type-preserving data augmentation, we used a benchmark classifier for MNIST digits, which achieves $99.54\%$ accuracy over 10,000 test samples[1]. Applying the imported classifier to the augmented test samples yields $96.14\%$ classification accuracy, which demonstrates that the augmenter preserves the label information (type) for $96.58\%$ of the augmented samples. During training of cpl-mixVAE, each agent received an augmented copy of the original image. To interpret the role of the continuous factor, we fix the discrete latent variable and change the state variable according to the conditional state distribution learned for each category. Fig. 2(b-e) illustrates these continuous latent traversal results for four dimensions of the state variable obtained by cpl-mixVAE. Each row corresponds to a different dimension of the categorical variable, and the state variable monotonically changes across columns. Panels (b), (c), and (d) represent commonly-identified continuous factors with global attributes, while panel (e) represents roundness, all in a digit-dependent manner.

Table 1 displays the classification performance of the discrete latent variable (as the predicted class label) for InfoGAN, different single-agent VAE methods including JointVAE and CascadeVAE, and cpl-mixVAE. We report the accuracy of the categorical assignments (ACC) and the reconstruction loss across 10 random initializations. For CascadeVAE, we used the numbers reported in (Jeong & Song, 2019). For InfoGAN and JointVAE, we used the publicly available implementation and training procedure reported in (Chen et al., 2016; Dupont, 2018). All reported numbers for cpl-mixVAE models are average accuracies calculated across both agents. We reported the performance of the proposed coupled VAE for four cases: (i) cpl-mixVAE($\mathbf{s} \not{\mid} \mathbf{c}$), in which the state variable is independent of the discrete variable; (ii) cpl-mixVAE($\mathbf{s} \mid \mathbf{c}$), which is the model in Eq. 6 and includes a pair of independent architecturally identical autoencoder agents using the proposed data augmentation in Eq. 7; (iii) cpl-mixVAE*($\mathbf{s} \mid \mathbf{c}$), which is also trained according to Eq. 6 and uses the proposed data augmentation, however in this setting, agents are not independent and the networks parameters are shared across agents; and (iv) cpl-mixVAE$^a$($\mathbf{s} \mid \mathbf{c}$), in which we used random rotation ($[-20, 20]$ degree) as an affine transformation for augmentation. Our results show that the cpl-mixVAE($\mathbf{s} \mid \mathbf{c}$) obtained the best categorical assignment among all models. Moreover, cpl-mixVAE$^a$($\mathbf{s} \mid \mathbf{c}$) also

---

[1]Digit Recognizer, kaggle competition: https://www.kaggle.com/c/digit-recognizer

| Method | $\mathcal{L}_{\text{rec}} \downarrow$ | ACC $\uparrow$ (mean $\pm$ s.d.) |
|---|---|---|
| InfoGAN | 169.4 | $77.87 \pm 09.54$ |
| CascadeVAE | - | $81.41 \pm 09.54$ |
| JointVAE | 122.0 | $68.99 \pm 11.76$ |
| JointVAE$^\dagger$ | 126.7 | $62.19 \pm 05.73$ |
| JointVAE$^\ddagger$ | 130.5 | $68.21 \pm 09.58$ |
| cpl-mixVAE($\mathbf{s} \not\mid \mathbf{c}$) | 113.9 | $79.63 \pm 08.32$ |
| cpl-mixVAE$^*$($\mathbf{s} \mid \mathbf{c}$) | 110.1 | $80.25 \pm 05.37$ |
| cpl-mixVAE$^a$($\mathbf{s} \mid \mathbf{c}$) | **105.9** | $82.92 \pm 04.64$ |
| cpl-mixVAE($\mathbf{s} \mid \mathbf{c}$) | 113.5 | $\mathbf{84.56 \pm 06.47}$ |

Table 1: Clustering results for the MNIST dataset, over 10 runs with $15K$ training iterations. For InfoGAN, we used the same network and the same parameters reported in the original paper by Chen et al. (2016). For CascadeVAE, all Joint-VAEs, and cpl-mixVAE models, we used $|\mathbf{c}| = 10$, $|\mathbf{s}| = 10$, and $\tau = 0.67$. For cpl-mixVAE models, the coupling factor is set to $\lambda = 1$.

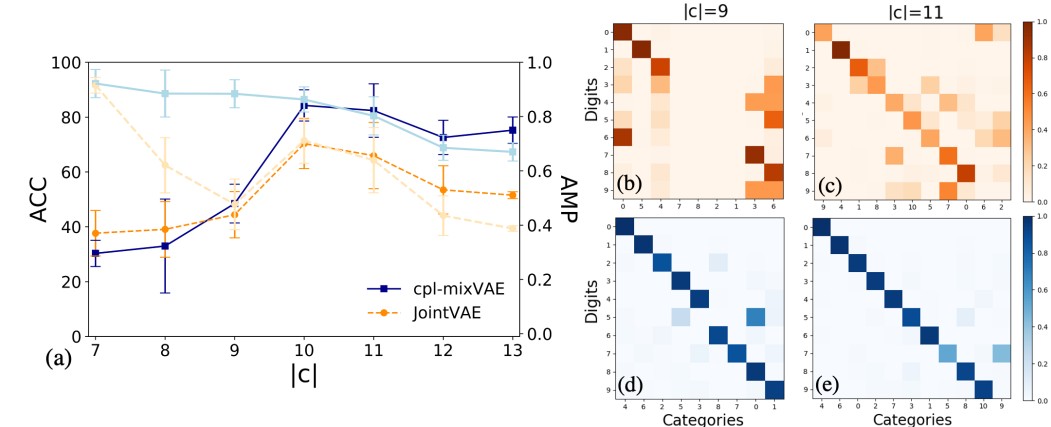

Figure 3: Clustering performance for the MNIST dataset, when the number of discrete variable ($|\mathbf{c}|$) is not equal to the true number of clusters (10 in this case). (a) ACC represents the accuracy of categorical assignment and AMP denotes average maximum posterior probability for JointVAE and $1^{st}$ agent of cpl-mixVAE. Error bars indicate mean $\pm$ s.d. over 5 randomly initialized runs. (b-c) Confusion matrices for JointVAE with AMP=0.57($|\mathbf{c}| = 9$) and AMP=0.54($|\mathbf{c}| = 11$); (d-e) Confusion matrices for $1^{st}$ agent cpl-mixVAE with AMP=0.88($|\mathbf{c}| = 9$) and AMP=0.80($|\mathbf{c}| = 11$). Color bar indicates per-category accuracy.

achieved the second highest performance, which demonstrates that even using a simple augmentation strategy can enhance the representation learning. For a fair comparison, Gumble-softmax temperature, $\tau$ and latent dimensionality are set to the same values as those for JointVAE and CascadeVAE. To understand whether architectural differences put JointVAE at a disadvantage, we report the results for JointVAE$^\dagger$, which uses the same architecture for the basic encoder/decoder networks as the one used in cpl-mixVAE. That is, JointVAE$^\dagger$ uses the same learning procedure as JointVAE, but its convolutional layers are replaced by fully-connected layers (see supplementary Section J for implementation details). Comparison of the results obtained with JointVAE and JointVAE$^\dagger$ suggests that the superiority of cpl-mixVAE is not due to the network architecture. Additionally, to separate the impact of augmentation in the training, we report the results for JointVAE$^\ddagger$, in which the JointVAE model has been trained with noisy copies of the original MNIST dataset generated by the proposed augmentation method. The reported clustering performance for JointVAE$^\ddagger$ suggests that including data augmentation by itself does not enhance the categorical assignment.

We also investigate the performance of cpl-mixVAE($\mathbf{s}|\mathbf{c}$) for different cardinalities of the categorical variable, $\mathbf{c}$. Fig. 3a shows the performance of cpl-mixVAE in terms of ACC and AMP as a function of $|\mathbf{c}| = K \in [7, 13]$. Here, AMP denotes the average of maximum posterior of categories i.e., $1/K \sum_{k=1}^{K} \max p(c_k|\mathbf{x})$. Expectedly, an insufficient number of categories results in inaccurate discrete variability encoding, which causes some dimensions being allocated to more than one digit (Fig. 3b and Fig. 3d), which results in lower ACC, but higher AMP. On the other hand, additional $c_k$ leaves some categories under-utilized (Fig. 3c and Fig. 3e), which does not only lead to lower ACC, but also lower AMP. Notably, our results show that while JointVAE suffers from sensitivity to empirical choices of $|\mathbf{c}|$ (Fig. 3b and Fig. 3c), cpl-mixVAE is more robust in encoding the discrete variability (Fig. 3d and Fig. 3e). As the AMP measure shows, for $|\mathbf{c}| < 10$, cpl-mixVAE utilizes all categories, without suffering from collapse and for $|\mathbf{c}| > 10$, it does not allocate unneeded categories and maintains high categorical assignment accuracy.

Table 2: Categorical assignment accuracies (ACC) and disentanglement scores (DS) for the dSprites dataset, over 10 randomly initialized runs with $|\mathbf{c}| = 3$, $|\mathbf{s}| = 6$, $\tau = 0.67$, and $\lambda = 10$.

|  | JointVAE | CascadeVAE | cpl-mixVAE($\mathbf{s} \mid \mathbf{c}$) |
|---|---|---|---|
| ACC (mean $\pm$ s.d.) | $44.79 \pm 03.88$ | $78.84 \pm 15.65$ | $\mathbf{96.30 \pm 09.15}$ |
| DS (mean $\pm$ s.d.) | $74.51 \pm 05.17$ | $\mathbf{90.49 \pm 05.28}$ | $89.98 \pm 04.09$ |

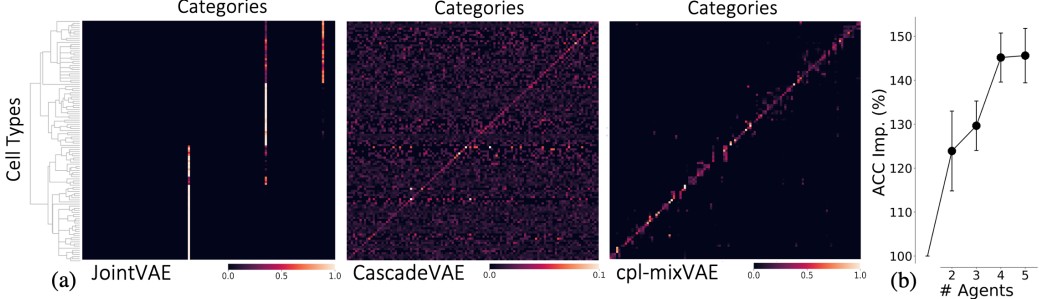

(a)   (b)   (c)

Figure 4: Interpretable continuous latent traversals of the trained cpl-mixVAE model with 6-dimensional continuous variable and 3-dimensional categorical variable for the dSprites dataset. Examples of (a) rotation, (b) scale, and (c) position are shown. The discrete variable is held as fixed for all reconstructions in the same row.

**dSprites:** In this dataset, due to the uniform distribution of classes, we again used a 2-agent cpl-mixVAE model. Similar to JointVAE and CascadeVAE, we used a 3-dimensional categorical variable for learning the shape (type), and a 6-dimensional state variable representing the style of each shape. Fig. 4 illustrates these traversal results for the three dimensions of the state variable obtained from the cpl-mixVAE. Each row corresponds to a different dimension of the categorical variable, and the state variable monotonically varies across columns. Table 2 shows the degree to which cpl-mixVAE outperforms the other methods in terms of categorical assignment accuracy. In addition, we reported disentanglement scores for all methods. For a fair comparison, we used the same disentanglement evaluation metric implemented for CascadeVAE (Jeong & Song, 2019).

**scRNA-seq:** Compared to typical machine learning datasets, the scRNA-seq dataset is exceedingly high-dimensional, with over $10,000$ genes. It includes $22,365$ neurons, over $100$ cell types with sizeable difference between the most and the least abundant clusters. Here, we excluded non-neuronal cells and used a subset of $5,000$ most expressed genes based on their peak values. While more than $115$ neuronal types are suggested for this dataset (Tasic et al., 2018), a significant challenge of representation learning in this dataset is its substantial imbalance, where for the most- and the least-abundant types, there exist $1404$ and $16$ samples, respectively. From the perspective of neuroscience, neurons as the basic building blocks of the brain, display both significant diversity and stereotype in their shapes, gene expression, and response patterns. Individual cells inherently differ due to either their type or continuous within-type variations (Trapnell, 2015; Andrews & Hemberg, 2018).

We used a 115-dimensional discrete and a 2-dimensional continuous variable for discrete and continuous neuronal factors representation, respectively. Fig. 5a illustrates the performance of JointVAE, CascadeVAE, and a 2-agent cpl-mixVAE model. The dendrograms on the y-axis displays the hierarchical relationship between neuron types according to Tasic et al. (2018). For many neuronal cells, whether the observed diversity corresponds to discrete variability or a continuum is an ongoing debate. While both JointVAE and CascadeVAE failed to identify meaningful cell types, cpl-mixVAE

Figure 5: Categorical assignments for the scRNA-seq dataset. (a) Confusion matrices of JointVAE, CascadeVAE, and cpl-mixVAE trained by $|\mathbf{c}| = 115$, $|\mathbf{s}| = 2$, and $\tau = 1$, over 45K iterations. For each model, hyperparameters were assigned as followings: $\mathcal{C}_s \in [0, 7]$, $\mathcal{C}_c \in [0, 10]$ (over $10K$ iterations), and $\beta_s = \beta_c = 100$ for JointVAE; $\beta \in [0.1, 10]$, and $\lambda = 0.1$ for CascadeVAE; $\lambda = 1$ for cpl-mixVAE. The dendrogram on the y-axis shows marker-based hierarchical classification with 115 cell types. (b) Accuracy improvement by adding more agents for cpl-mixVAE, over 3 runs. $A$-agent's performance for $A \geq 2$ is compared with a baseline 1-agent, JointVAE.

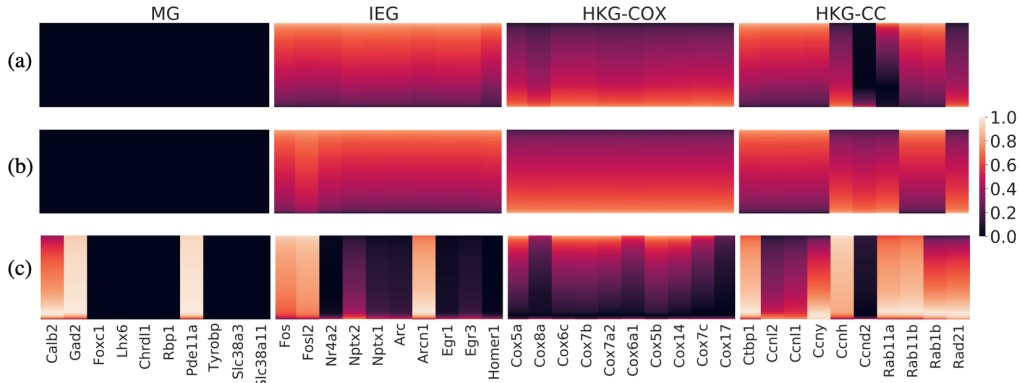

Figure 6: Continuous latent traversals for two excitatory cell types (a) "L5 NP ALM Trhr Nefl" and (b) "L6 CT Nxph2 Sla", and an inhibitory cell type (c) "Pvalb Akr1c18 Ntf3". For each type, the continuous latent traversal is color-mapped to a normalized reconstructed gene expression value (colorbar) as a function of the state variable for four gene subsets from left to right: marker genes (MG), immediate early genes (IEG), and two subgroups of house keeping genes, cytochrome c oxidase (HKG-COX) and cell cycle regulators (HKG-CC).

successfully identified the majority of known types. The, accuracy for categorical assignment across the entire 115 types is 39% (chance level is ∼ 6%, based on the most abundant type). Unlike the discussed benchmark datasets, the neuronal types are not uniformly distributed, Accordingly, as another experiment we applied more than two agents on the scRNA-seq dataset to investigate the accuracy improvement for categorical assignment. Fig. 5b illustrates the accuracy improvement with respect the a single agent model, i.e. JointVAE.

To examine the role of the continuous latent variable, which can profile activity-regulated genes, we applied a similar traversal analysis to that used for the MNIST and dSprites datasets. For a given cell sample and its discrete type, we changed every dimension of the continuous variable using conditional distribution, and inspected gene expression changes caused by continuous variable alterations. Fig. 6 shows the results of the state traversal experiment for two excitatory neurons belonging to the "L5 NP" (near-projecting) and "L6 CT" (corticothalamic) sub-classes, and an inhibitory neuron belonging to the "PV" (parvalbumin) class. In each sub-figure, the latent traversal is color-mapped to normalized reconstructed expression values, where the y-axis corresponds to one dimension of the continuous variable, and the x-axis corresponds to four gene subsets, namely (i) marker genes (MG) for the two excitatory types, (ii) immediate early genes (IEG), and two house keeping gene (HKG) subgroups (iii) cytochrome c oxidase (COX), and (iv) cell cycle (CC) regulators (Hrvatin et al., 2018; Tarasenko et al., 2017). MGs are normally expected to function as indicators for particular cell types whose normalized expression is unaffected by the regulatory activities of the cell. Indeed, the expression of the reported excitatory MGs remains constant for excitatory traversals but not necessarily for the inhibitory traversal (i.e., Calb2, Gad2, Pde11a in Fig. 6). In contrast, the expression of IEGs and HKGs should depend strongly on the metabolic and environmental conditions. Indeed, we find that the expression changes of IEGs and HKGs are for the most part monotonically linked to the continuous variable, reaffirming that the it captures relevant, interpretable continuous variability, as in the MNIST and dSprites examples. Lastly, the expression of IEGs and HKGs (activity-regulated genes) depends on the cell type e.g., not all IEGs are activated for all cell types. Notably, for the excitatory "L5 NP" and "L6 CT" cells that are proximate in the hierarchy (Tasic et al., 2018), state traversal is quite similar. These results suggest that the continuous variable inferred by cpl-mixVAE provides insight when deciphering the molecular mechanisms shaping the landscape of biological states, e.g. metabolic, disease.

## 6 CONCLUSION

We have proposed cpl-mixVAE as a multi-agent framework using a type-preserving data augmentation to apply the power of collective decision making in unsupervised joint learning of discrete and continuous factors. This framework utilizes multiple pairwise-coupled autoencoding agents with a shared categorical variable, while independently learning the continuous variables. Our experimental results for all three datasets show that cpl-mixVAE outperforms comparable models. In addition, for a challenging gene expression dataset, we showed that the proposed framework can identify annotated neuronal types and differentiate between type-dependent and activity-regulated genes.

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
