# OpenReview forum: "Mixture Representation Learning with Coupled Autoencoding Agents"
_ICLR.cc/2021/Conference — Reject_

### Official Review · AnonReviewer4 · 2020-10-26

**Rating:** 6
**Confidence:** 5

**Review:**

====================================================================================================

Summary :

The paper proposed the new disentanglement approach based on the "wisdom of the crowd". First, the proposed method enforces the consensus on the categorical assignments from the different agents. Each agent receives a similar image that is generated from the data augmentation method (based on the VAE-GAN technique).  The encoder of each agent first estimates the categorical distribution and estimates continuous variables from the categorical data and original image. The proposed method focuses on disentangling categorical information. The evaluation also focuses on categorical data and the proposed method outperforms the other single-agent baselines.

===================================================================================================

Reason for score:

Overall, I'm on the borderline between acceptance and rejection. The proposed method is persuasive, but the ablation study is required to improve this paper. I will finalize my decision after reading the reviews from the other reviewers and authors' responses to my concerns.

====================================================================================================

Strong points:

(1) I was impressed by the idea to generate a similar image with the same class label through techniques developed from VAE-GAN since this idea reverts the unsupervised setting into a weakly-supervised setting. In this regard, I have a concern (See cons).

(2) I like the proposed multi-agent approach to increase the robustness of the classification label. Maybe this method largely reduces the variance from the different seeds.

====================================================================================================

Cons :

The idea which leverages a similar image with the same class label is quite similar in the disentanglement under weakly supervised setting as in [1]. Therefore, type-preserving augmentation seems critical in the performance of the proposed method. In this sense, I believe the ablation study on the augmentation method is required to persuade readers and to provide guidelines for future research. (e.g. Adding gaussian noise, Mirror image)

Even though the disentanglement score cannot be evaluated without supervision on the labels, I recommend authors to compare the disentanglement score with other baselines(Joint VAE, Cascade VAE), 'I(s;c)', and etc.

=============================================================================================================

After Rebuttal :

Sorry for the delay. I checked the comments of the other reviewers, responses, and the revised version. The authors address my concerns in the revised version. I vote for marginal acceptance.

[1] Weakly-Supervised Disentanglement Without Compromises, ICML 2020

---

> ### Author Response · Authors · 2020-11-21
> **Response to reviewer 4**
>
> We revised the manuscript and included a Related Work section, where we cite and discuss the paper suggested by the referee.
>
> -To address the reviewer’s comment on ablation studies on the augmenter:
>
> We have included more assessment for the proposed data augmentation for both image and non-image datasets (Experiments section and the supplement of the revised manuscript, Section I). Our quantitative study on the MNIST dataset reveals that the augmenter achieves ~96.5% type-consistency. Please also see a similar question by AnonReviewer1.
>
> We also added new theoretical analysis on the consequences of the augmenter potentially under-exploring the variability within a cluster. Briefly, the Remark after the proof of Proposition 1 shows that the multi-agent framework will still be useful in this scenario, except that each agent will contribute less than before to improving the confidence score.
>
> In addition, we would like to bring the reviewer’s attention to two experimental studies in the Table 1:
>          (i) evaluating a single agent model performance using the proposed data augmentation approach, in which we showed that adding the augmenter only slightly degrades the performance of JointVAE (JointVAE++ in Table 1). This is due to the 3.5% performance gap. (mentioned above)
>          (ii) We have also reported a study on the cpl-mixVAE model by replacing the proposed VAE-GAN-like augmenter with an augmenter that applies a set of conventional image augmentations including affine transformation to the MNIST image (reported as cpl-mixVAE^a in Table 1). This study shows that this affine augmenter performs slightly worse than the proposed VAE-GAN-like augmenter.
>
>
> -Per the reviewer’s request, we have now added the disentanglement score to Table 2, for the dSprite dataset. To have a fair comparison with the alternative methods, we used the original disentanglement score reported for CascadeVAE and JointVAE. Briefly, our method jointly achieved high accuracy and disentanglement scores. On the other hand, we would like to point out that in calculation of this score (suggested by Kim et al., 2018), it is assumed that all latent factors are independent (disentangled). However, in the proposed cpl-mixVAE model, the continuous factor can depend on the discrete random variable to represent class-dependent variabilities. Since the dSprites is a simulated dataset, in which the continuous factors like scale, rotation, etc., are independent of the geometrical shape (class label), the model can infer q(s|c,x) as q(s|x). Therefore, we can compute the disentanglement score over all factors, the same as cascadeVAE. But for real-world applications, the continuous factor often depends on class label and the suggested disentanglement score is not a proper measure to assess the mixture representation. For instance, the scRNAseq experiment shows that the continuous factors depend strongly on the categorical variable (Fig. 6), and our method significantly outperforms cascadeVAE on this complicated, highly imbalanced problem (updated Fig. 5).

---

### Official Review · AnonReviewer1 · 2020-10-27
**The authors propose a framework to learn a mixture distribution (a distribution which consists of continuous and categorical variables) using a coupled variational autoencoder framework. The different autoencoders are trained using augmented samples which are generated using a VAE-GAN framework. The idea of using coupled autoencoders is compelling and it makes sense and is well-known that data augmentation increases performance - given the ‘new’ data is good enough.**

**Rating:** 5
**Confidence:** 3

**Review:**

Quality
The paper proposes good ideas that are compelling and make sense. Using multiple agents and data augmentation seems directions worth pursuing to improve mixture representation learning.
The experiments are mostly clear and well-motivated. Unfortunately, the related work is limited and could be improved and extended. For the evaluation of the data augmentation step, there is only a qualitative evaluation in Figure 2. A more in-depth analysis regarding the type-consistency would strengthen this part of the pipeline.
In section 2, the authors say that previous work falls short in efficiently learning mixture distributions due to hyper-parameter tuning or additional cost due to the optimization procedure. In the proposed work, there are some (e.g. coupling hyper-parameter lambda, temperature tau or gamma in data augmentation) new hyper parameters introduced as well. This somehow contradicts the motivation of the method and it is not evident why the proposed method is more efficient. Additionally, ablation studies on critical hyper-parameters like lambda or the temperature parameter would be interesting to see and strengthen the paper. Without further evidence, the claim of a more efficient method is not really justified in my opinion.

Clarity
In general, the paper is well and clearly written. A related work section is missing which makes it more difficult to position the proposed work with respect to previous work. The authors motivate the representation learning part but sometimes use the term clustering. In my opinion, this weakens the motivation and goal of the paper.

Originality
The paper presents a novel view on mixture representation learning using data augmentation which itself is not a new idea nor do they use a new method to augment the data. The multi-agent view on mixture representation learning is a new idea (to my knowledge, but not aware of all related work) making the paper original.

Significance
The ideas of the paper are significant and worth pursuing.

Questions to the authors
-Proof of Proposition 1, equation 7: I do not fully understand where the factor A is coming from in eq. 7. Isn’t this assuming that all agents agree on p(x_a | phi = m)? Is this valid? Thanks for explaining this a bit more in detail.
-Is your final objective still a valid ELBO? In the proof in the appendix you use approximations to derive the objective, but not only bounds. So I am not sure if it is still an ELBO.
-In the experiments (Table 1), why is only jointVAE used with augmented data and not cascadeVAE which seems to perform better in its vanilla version?

Further Comments
-The work has limited comparison to previous work. Bouchacourt et al.’s “Multi-Level Variational Autoencoder: Learning Disentangled Representations from Grouped Observations” presents a similar idea of using content/class and style/state spaces (only with continuous variables). A comparison to this work showing the potential benefit of using a discrete class space would improve the quality of the paper.
-I am happy to upgrade my score if the authors address my concerns.

---

> ### Author Response · Authors · 2020-11-21
> **Response to reviewer 1**
>
> We now revised our manuscript and added a Related Work section, which discusses papers beyond those already cited in Introduction, including the paper suggested by this reviewer.
> - We have included additional quantitative and qualitative evaluation for the data augmentation (Experiments section and Section. Our analysis on MNIST suggests that the augmenter achieves ~96.5% type-consistency. Briefly, we imported a state-of-the-art classifier for MNIST achieving ~99.5% classification accuracy. We passed the augmented test samples to this classifier to obtain the result reported above. We revised our manuscript to include this new analysis. Moreover, to demonstrate the performance of the data augmentation for the non-image dataset, we reported the augmentation results for the single cell data set including the genes expression profiles (revised supplement, Section I).
>
> - To address the reviewer’s comment on the hyperparameters of the proposed cpl-mixVAE model, we would like to point out:
>
> (i) Requiring a fine parameter tuning process is one of multiple reasons for the alternative methods in falling short. Results across different datasets demonstrate some of those issues, e.g. mode collapse. We observed that alternative methods depended strongly on parameters tuning. One of the main challenges of methods like JointVAE is their reliance on channel capacity parameters, which need to be dynamically adjusted throughout training in a nontrivial way, and independently adjusted for each discrete and continuous variable, while there could be dependency between those factors. JointVAE requires three parameters just for each latent space, i.e. minimum capacity, maximum capacity, and the number of iterations to change the capacity. In contrast, the proposed cpl-mixVAE only requires a coupling factor (lambda) that regularizes the cooperation across agents. For all datasets, we suggested 1=<lambda<=10. In the data augmentation implementation, we also used the same gamma value across all datasets, without any additional fine tuning.
> We now added a new section in the supplement(Section H), in which we investigated the sensitivity of the cpl-mixVAE performance to lambda, in comparison with JointVAE with four critical parameters. Our results show that while cpl-mixVAE's performance remains robust within a window covering two orders of magnitude, JointVAE is sensitive to changes to the channel capacity.
>
> (ii) The temperature parameter (tau) is not an additional parameter in our framework. It is a parameter of the Gumbel-softmax distribution which is part of both alternative methods, JointVAE and CascadeVAE. In all our experiments, we used the same temperature values suggested by the previous studies, without any tuning. The study by Jang et al., 2016 discusses the impact of tau on the discrete probability estimation.
>
> - The proposed multi-agent framework assumes only that the agents agree in expectation on the probabilities, and not on a point-by-point basis: the agents have identical architectures, they are i.i.d. initialized, and our training method is completely symmetric over the agents. To avoid further confusion, we have revised the proof of Prop. 1 and clarified these aspects in both main text and the supplement. (Please also see a similar question by Reviewer3)
>
> - The approximation used in the ELBO derivation for the joint discrete distribution, is using the Midpoint rule.  We now added new analysis bounding the error in calculating joint categorical distribution using the Midpoint rule. To show the quality of the ELBO-approximation, we explicitly formulated the error term and showed the error bound on the joint probability is a function of a constant value, independent of the variational inference optimization parameters.
>
> - Type-preserving augmentation is not helpful for 1-agent framework, because, (i) we are not using data augmentation to increase the training sample size; (ii) our augmentation method is ~96.5% type-preserving. Therefore, utilizing augmented samples is helpful only when there are multiple agents cooperating to make a decision. This is demonstrated by JointVAE performing slightly worse when it uses augmented samples as well.
> On a related note, we have now added new analysis with CascadeVAE on the scRNAseq dataset (Fig. 5), which shows CascadeVAE performs significantly worse than our method, while it is ~70 times slower than our method on the same GPU. This can be attributed to both the complexity of the dataset and the strong reliance of CascadeVAE  assumption on a uniform categorical prior (Lemma 1 in the CascadeVAE paper).
>
> - We now discuss the ML-VAE approach in the new Related Work section. The ML-VAE method proposed by Bouchacourt et al. is a weakly supervised disentanglement method, in which class label information is used for grouping samples. In contrast, here we focus on the fully unsupervised representation learning problem and without relying on any class label information.

---

### Official Review · AnonReviewer2 · 2020-10-27
**OK submission but lack of  motivation and proposed model is computationally expensive**

**Rating:** 5
**Confidence:** 3

**Review:**

This paper proposed a multi-agent VAE model that combines multiple copies of VAEs with coupling constraints to improve its latent representation learning (by encouraging discrete variable consistency). The experiments show that the proposed model outperforms other discrete&continuous VAE models in terms of clustering ACC. The experiment is conducted on MNIST and scRNA-seq datasets.

Overall
===
I think this is a good submission in terms of describing its methodology. However, it is hard to justify its motivation as the model is over-complex, and the same task could be achieved by other approaches such as deep clustering. It is better if the authors could include such discussions in the paper.

Pros
===
1. This paper gives sufficient justification on why we need multiple copies of estimators to reach better estimation.
2. The method description given in Section 3.1 is quite clear as the equations are self-explainable.
3. Results shown in table 2 indicate significant performance improvement.


Cons
===
1. The approach proposed itself is very complex comparing to other generative model architecture. It is hard to justify if the model can generalize to other tasks other than simple, low-dimensional tasks. Consider maintaining multiple VAEs graphs in memory; it appears hard to take advantage of this work. Is the cost of deploying this model worth the performance improvement (as Table 1 shows limited improvement)? Also, to train such a model, the user also needs to produce a type-preserving augmentation, which is very costly for clustering.

2. The proposed model jointly optimizes the main objective (as Equation 3) and also optimizes the relaxed equivalent constraints at the same time (as Equation 6 last component). Is there any justification why not alternative optimization but joint? As the DeepClustering paper mentioned, alternative optimization is better than joint in their case. Not sure if there will be a similar observation here. Isn't it comparable to the DeepClustering paper if the goal is to do the clustering and interpretation?

3. The introduction of this paper gives me a hard time to follow as the terminology used is uncommon in generative model literature (single-agent, multi-agent). I was confused as it appears to be an RL paper at the beginning.


Minors
===
1. Which style state shown in Figure 2 (b-e) if you have multiple agents? The description said those are four style/state dimensions, but there are two agents in the experiment setting, and each of them has 10 style/state dimensions. Please be more precise.
2. Figure 1 (b-c) is not quite informative, and they are too close to each other. It is better to provide a more intuitive demonstrative figure.

---

> ### Author Response · Authors · 2020-11-21
> **Response to reviewer 2**
>
> We would like to begin by addressing a potential confusion about the main objective of the paper: we have proposed a general framework for the joint representation learning problem, whose aim is to learn interpretable variational factors. Our results demonstrate the ability to build consensus clustering in an online fashion jointly with dissection of interpretable continuous factors. In contrast, it is not clear how the approach suggested by the reviewer (see below for a detailed discussion on “Deepcluster” and alternating optimization) is immediately applicable to this problem.
>
> - We would like to bring the reviewer's attention to a key point that although the proposed cpl-mixVAE framework is using a type-preserving data augmentation module and multiple independent autoencoders, the complexity of the entire framework in terms of optimization run time, number of hyperparameters and required prior knowledge/assumption is less than that of alternative methods. For instance,
> (a) the training run time for CascadeVAE method is roughly 70 times longer (on the same GPU hardware) than our method due to its external optimization, based on the RNAseq experiment.
> (b) For each dataset, JointVAE requires the tuning of hyperparameters (which are sensitive) which is computationally expensive and challenging in practice.
>
> - To support the practicality of our method on multiple applications with different complexities, we showcased its performance not only on benchmark datasets including MNIST and dSprites, but also on a highly non-uniform single cell dataset with ~100 clusters (Fig. 5 and 6 in the main manuscript). We note that none of the benchmark datasets, e.g. MNIST, CelebA, dSprites, have more than 10 clusters, nor inherent class imbalances.
> In the revised manuscript, we have updated Fig. 5, which now shows both CascadeVAE and JointVAE performing significantly worse than our method. Moreover, we added an additional section in the supplement (Section H in the updated manuscript) to demonstrate the necessity of parameter tuning for JointVAE.
>
> - In the second point under ‘Cons’, the reviewer mentioned ‘the DeepClustering paper’. While a web search for ‘DeepClustering’ or ‘Deep Clustering’ produces multiple potential papers, including the highly popular paper ‘Deep Clustering for Unsupervised Learning of Visual Features’ by Caron et al, to the best of our understanding, the reviewer is referring to ‘DeepCluster: A General Clustering Framework based on Deep Learning’ by Tian et al., 2017.
>
> To address the reviewer’s comment:
> (i) the proposed “DeepCluster” approached by Tian et al. is a clustering algorithm and cannot be utilized for the joint mixture representation learning problem, which is our main focus in this study.
> (ii) cascadeVAE’s approach is a similar approach to what the reviewer suggested, which uses alternating optimization. The study on the RNAseq data shows it performs significantly slowly and significantly worse (updated Fig. 5).
> While trying to use ADMM sounds like an interesting future direction, it is not immediately applicable to our model because of our joint continuous factor dissection and consensus building goals.
> (iii) We cited Tian et al. , 2017 in the new Related Work section in a comparative way.
>
> - To address the third comment and prevent further confusion, we have revised the manuscript and explicitly defined the “multi-agent autoencoder framework” (Definition 1 in the revised manuscript) by introducing autoencoding units as ‘agents’. In this framework, since autoencoding agents are independent machines which are cooperating with each other during training only and only through the cost function, we called it a “multi-agent framework”.
>
> - To address the minor comments:
> The reported results in Fig. 2 were obtained from one of the agents of the cpl-mixVAE framework with two autoencoding agents, where each agent mixture representation is parameterized with 10-dimensional continuous and 10-dimensional categorical variables. We have revised the figure caption and included the missing information.
>
> We have revised Fig. 1 to provide a better illustration for our framework.
>
> We respectfully hope that we were able to address any confusions, and the reviewer will revise their initial score in light of these clarifications.

---

### Official Review · AnonReviewer3 · 2020-11-02
**Interesting work, but some concerns with the current presentation**

**Rating:** 6
**Confidence:** 3

**Review:**

This work presents a new approach to handling categorical latent variables in VAEs. The method has two key components: a *multi-agent* architecture in which categorical assignments are generated through consensus across multiple VAE models and a data-augmentation method which allows each model (termed *agent*) to be trained on a perturbed version of the original data. The authors provide some theoretical justification for their approach and evaluate on two benchm`ark data-sets and one real-world application.

**Strengths**
1. The method presented seems to be quite novel, with various technical contributions required to jointly train the separate VAE models in a way that they didn't collapse to a single model.
2. The evaluation on MNIST and dSprites provide empirical evidence that this method out-performs a number of baseline methods. To my knowledge, the methods selected are good candidates for being SOTA methods for VAEs with discrete latent variables.
3. The application to the scRNA-seq provides further evidence that the method works well on real-world data.

However, I have some concerns with the presentation of the paper as is. If the authors can address these, I am happy to increase my score.

**Questions / Concerns**
1. While I did not have time to check all the proofs in detail, I wasn't convinced by the proof of Proposition 1 in Appendix A. A few specific things on which I would appreciate clarification from the authors:

a) It seems like the proof (as stated) would follow if all the agents were identical, but this doesn't match what one would expect. Can the authors make the impact of using different agents more explicit in the assumptions and explain how this leads to the conclusion?

b) The proof assumes the augmented samples are generated from p(x | \phi = n) but the method generates augmented samples as augmentations of a specific training example x_n. Can the authors explain why they believe this isn't an issue to the relevance of their result?

c) Going from (6) to (7) appears to assume that the probability for each augmented sample is the same. We would not expect this to be the case for a single agent or multiple agents. Can the authors explain the basis for this step?

2. I believe there are a number of different lines of research which could be considered as relevant related work which the authors have missed.

a) Boosting Variational Inference [1,2] describes techniques which use a mixture of variational distributions within the VI setup to provide a better approximation to the posterior. While the method presented here is not strictly a boosting method, it is quite possible that the gains seen are a result of being able to better approximate the true posterior by allowing the model to fit multiple inference networks.

b) Consensus clustering aggregates the results of a clustering algorithm over multiple initialisations [3]. This often gives an improvement over the output of a single instance of a clustering algorithm and can be applied post-hoc to any algorithm that outputs cluster assignment. For example, a consensus version of CascadeVAE could be considered as an additional method in the experiments.

c) Co-training is a method for training classifiers on multi-view data such that they predict same labels for co-occurring patterns in each view. This was originally presented by Blum and Mitchell [4] as an approach for semi-supervised learning and was subsequently applied to unsupervised learning by Kumar and Daume III [5]. The key idea in methods that leverage co-tr`aining is to use consensus across the different models, which feels similar to the consensus constraint in equation (3). As well as being relevant related work, the theoretical perpsective of co-training might also be useful to address some of the concerns stated about the theoretical result in this paper.

3. I'm not sure how informative Figure 3 is. It's to be expected that the best performance will come when the dimension of the latent variable is equal to the true number of classes, but this will not be available in the fully-unsupervised case where the true classes are unknown. It would be useful if model selection using AMP was shown to lead to the correct number of classes for MNIST, but it wasn't clear to me if this could be inferred from the plot. If this is what the authors intended to show, they should add some additional text to explain why this is the case.

4. In the experiments on MNIST, the authors compare to the case where m=10, S=10 (Table 3 of Jeong and Song). But Jeong and Song reported higher accuracy (with lower variance) for  the case with m=4, S=10. Why did the authors choose to compare against the lower performing of the two configurations studied in Jeong and Song?

5. It wasn't 100% clear to me whether the model uses fully independent VAEs (i.e. separate parameters for encoder and decoder in each model), but I believe this is the case. It would be informative to compare against the case where the decoder network has the same parameters in each model. I suspect the gain is primarily due to the fact that we are aggregating over different encoders / inference networks and this additional experiment would make this clear.

**References**
1. Guo et al. (2016). Boosting Variational Inference
2. Locatello et al. (2018). Boosting Black Box Variational Inference
3. Monti et al. (2003). Consensus Clustering: A Resampling-Based Method for Class Discovery and Visualization of Gene Expression Microarray Data
4. Blum and Mitchell (1998). Combining labeled and unlabeled data with co-training
5. Kumar and Daume III (2011). A co-training approach for multi-view spectral clustering

---

> ### Author Response · Authors · 2020-11-21
> **Response to reviewer 3**
>
> 1) We thank the reviewer for drawing our attention to the proof of Prop. 1. We found typos in the proof, in our original submission. We have now corrected these typos, and revised the notations, presentation of the proof, and the main text surrounding Prop. 1.
>
> (a) The proposed multi-agent framework introduced in Prop. 1 assumes only that the autoencoder agents are identical in expectation, and not on a point-by-point basis: the agents have identical architectures, they are i.i.d. initialized, and our training method is completely symmetric over the agents. As suggested by the reviewer, we edited the proof to make the connection between the proposition and the implementation more precise and explicit (Eqs. 1,2,6 and surrounding text in the proof). We also explicitly define the multi-agent framework in the revised main text (Definition 1).
> Finally, we would like to add that Prop. 1 is not meant to capture all the details of the implementation closely, rather to motivate the general multi-agent framework. (Please also see a similar question by Reviewer1)
>
> (b) Our framework assumes that the augmentations ‘of a specific training example x_n’ are type-preserving. Therefore, the proposition/proof and the framework agree with each other in this regard. We have now explicitly tested, on MNIST, the validity of the type-preserving assumption, and found that our augmenter is ~96.5% type-preserving. We added this new result to the main text.
> If the reviewer is questioning the under-exploration scenario, in which augmented samples are concentrated around the given sample, the proof follows in the same way except the augmented samples are no longer conditionally independent. This means that each augmented sample adds less than before to the confidence score. Yet, the same argument shows that there will be an A for which the claims of the proposition are satisfied. We have added a Remark after the proof summarizing this discussion.
>
> (c) Going from Eq. 6 to Eq. 7 in the original submission, we do not assume the probabilities are the same. Rather, the equality is on the expectations of those probabilities. Specifically, we assume E_x[p(x_1|phi)] = E_x[p(x_A|phi)], which is justified when x_1, … x_A ~ p(x|phi).
>
> 2) We thank the reviewer for these suggestions. Indeed, we found them relevant to our work and we now cite these papers in an additional Section called Related work (in the updated manuscript) explaining how they relate to and differ from our work.
> Regarding consensus clustering, while it provides an interesting aspect as discussed in the new Related work section, it is highly unlikely to improve results when only 2 agents are used, because consensus clustering suggests a voting-like scheme. More importantly, we note that a novelty of our approach is to seek consensus online, at the time of training, not after the agents converge to their individual decisions. This influences the inference of the continuous factors because they depend on the categorical variable, which can only be addressed in a joint learning framework.
>
> 3) We thank the reviewer for pointing out this issue and apologize for not being clear about Fig. 3. The main goal of Fig. 3 is to study the performance of cpl-mixVAE for different cardinalities of the categorical variable, |c|. In the figure, we report two measures; ACC (categorical assignment accuracy) and AMP (average of maximum posterior of categories), where each of them represents different aspects of the cpl-mixVAE's performance in encoding the discrete variabilities. The main reason for reporting the AMP measure in this figure is to show that (i) for |c|<10, cpl-mixVAE utilizes all categories, without suffering from collapse, and (ii) for |c|>10, it does not allocate unneeded categories in the interest of high categorical assignment accuracy.
> As the reviewer correctly mentioned, since both measures utilize the class label, they cannot be used for finding the true number of clusters. To fix this point, we have updated Fig. 3 and its description to clarify the purpose of those plots.
>
> 4) Using a 10-dim continuous variable is quite standard in mixture representation learning for the MNIST dataset. JointVAE (Dupont, 2018), InfoGAN (Chen et al., 2016), and ML-VAE (Bouchacourt et al., 2018) all use 10-dim continuous variables. Indeed, the cascadeVAE paper also chose m=10 to study continuous variability and latent traversals on MNIST. They only claim that m=4 results in a higher clustering accuracy in isolation. It is, however, key to consider both discrete and continuous factors jointly, which is the main focus of our work.
>
> 5) We thank the reviewer for this suggestion. We have now performed this experiment, and revised Table 1 to report it, i.e. cpl-mixVAE*. Briefly, sharing parameters across agents produced competitive results. Yet, they were slightly worse than those of fully independent agents, which allow for better exploration of the parameter space.

---

> > ### Comment · AnonReviewer3 · 2020-11-24
> > **Updated Review**
> >
> > Thanks for your response and the updated version of the paper. I've increased my score to 6.

---

> > > ### Author Response · Authors · 2020-11-25
> > > **Further question(s)**
> > >
> > > We thank the reviewer for updating the score. If there are other concerns/questions, we would be happy to address them. We are quite excited about our work.

---

### Author Response · Authors · 2020-11-21
**General response**

The reviews were thorough and constructive. There was a clear consensus that our method is “quite novel” and “persuasive”, and our ideas are “compelling”, “significant”, “impressive”, and “original”. The reviews did not identify common weaknesses. Instead, each reviewer questioned different aspects of our work and asked for clarifications/experiments.
We believe that there is a mismatch between the qualitative consensus assessment and the quantitative scores – multiple reviewers explicitly mentioned that they’d be happy to increase their scores if they find our responses satisfactory.

Below, we address the concerns raised by the reviewers. We respectfully hope that these clarifications and analyses will warrant substantially higher scores.

Our approach achieves strong results on multiple datasets, outperforming state-of-the-art baselines. Yet, beyond quantitative results, we think that the conceptual advance is a key merit of our approach: we demonstrate online consensus building jointly with dissection of learning interpretable continuous factors, without assuming any supervision. We demonstrate that this approach can be implemented in a fully neural architecture, without calling external optimization algorithms. We have now updated our manuscript to clarify the presentation and add further analyses, as suggested by the reviewers.

Summary of main updates:
1) clarification and improvement of the presentation for Proposition 1
2) additional theoretical results on type-preserving augmentation (updated supplement, Remark 1)
3) a new section on related work (updated manuscript, Section 2)
4) quantitative and additional qualitative analyses on the type-preserving augmenter
5) a new result on the single cell RNA-seq dataset with cascadeVAE (updated manuscript, Fig. 5) to show that it performs significantly worse than the proposed cpl-mixVAE model
6) a new study on the robustness against hyperparameter tuning (updated supplement, Fig. 3)
7) providing the error bound for ELBO approximation (updated supplement, Section D)

---

### Author Response · Authors · 2020-11-25
**manuscript and supplement are updated**

We would like to thank all the reviewers for their valuable and constructive comments. To find our latest modifications, please see the newest manuscript and supplement files.

---

### Decision · Program_Chairs · 2021-01-07
**Final Decision**

**Decision:**

Reject

**Comment:**

This paper introduces and analyses a method to train a population of VAEs with mixed continuous (referred to as "style") and discrete (referred to as "labels") latent-variables. The population is trained under the constraint that inferred discrete latent variables  to be the same for all models.
The paper also investigates a data augmentation mechanism inspired by  (Antoniou et al., 2017).
The presentation is overall clear and the idea is interesting, although the language of "agents" is not standard in generative model literature and is a bit confusing. The experiments also show very good clustering results of the proposed method.
Unfortunately the pipeline was determined to be quite complex while the motivation for its design choices were unclear. This, combined with multiple concerns about the experimental validation, led to a reject decision.